# Pathophysiology and Main Molecular Mechanisms of Urinary Stone Formation and Recurrence

**DOI:** 10.3390/ijms25053075

**Published:** 2024-03-06

**Authors:** Flavia Tamborino, Rossella Cicchetti, Marco Mascitti, Giulio Litterio, Angelo Orsini, Simone Ferretti, Martina Basconi, Antonio De Palma, Matteo Ferro, Michele Marchioni, Luigi Schips

**Affiliations:** 1Department of Medical Oral and Biotechnological Science, Università degli Studi “G. d’Annunzio” of Chieti, 66100 Chieti, Italy; flavia.tamborino96@gmail.com (F.T.); rossella.cicchetti3@gmail.com (R.C.); m.mascitti@gmail.com (M.M.); giulio.litterio@gmail.com (G.L.); orsini.ang@gmail.com (A.O.); ferretti.smn@gmail.com (S.F.); basconimartina1@gmail.com (M.B.); antonio.depalma.93@gmail.com (A.D.P.); luigi.schips@unich.it (L.S.); 2Division of Urology, European Institute of Oncology, Istituto di Ricovero e Cura a Carattere Scientifico (IRCCS), 20141 Milan, Italy; matteo.ferro@ieo.it

**Keywords:** nephrolithiasis, kidney stones, calcium oxalate

## Abstract

Kidney stone disease (KSD) is one of the most common urological diseases. The incidence of kidney stones has increased dramatically in the last few decades. Kidney stones are mineral deposits in the calyces or the pelvis, free or attached to the renal papillae. They contain crystals and organic components, and they are made when urine is supersaturated with minerals. Calcium-containing stones are the most common, with calcium oxalate as the main component of most stones. However, many of these form on a calcium phosphate matrix called Randall’s plaque, which is found on the surface of the kidney papilla. The etiology is multifactorial, and the recurrence rate is as high as 50% within 5 years after the first stone onset. There is a great need for recurrence prevention that requires a better understanding of the mechanisms involved in stone formation to facilitate the development of more effective drugs. This review aims to understand the pathophysiology and the main molecular mechanisms known to date to prevent recurrences, which requires behavioral and nutritional interventions, as well as pharmacological treatments that are specific to the type of stone.

## 1. Introduction

Nephrolithiasis is an increasingly common disease with a prevalence of 2–15% worldwide [1]. Country-based differences in nephrolithiasis prevalence have been noticed. More specifically nephrolithiasis rates are about 13% in North America, 5–9% in Europe, and 1–5% in Asia [2]. Nephrolithiasis incidence has increased dramatically over the past 30 years due to environmental changes, including inadequate diet and restricted physical activity [3]. 

The prevalence of kidney stone disease is generally higher in males than females [4]. However, nephrolithiasis is becoming more and more common in women than men causing a reduction of the male to female ratio [5] as proved by the increasing number of annual visits for urinary stones in women at emergency departments [6]. 

Nephrolithiasis has an important economic burden on healthcare systems, with a positive outlook expected in the future due to the increasing incidence [5]. 

Such changes could be also explained by dietary changes. It is known that a diet rich in animal proteins and salts and low in fiber and plant-based proteins is associated with a low calcium content and increases the risk of lithogenesis. At the same time, low fluid intake or dietary changes (e.g., the wide spread of high protein diets or higher fructose intake), can also increase the risk of nephrolithiasis [6]. 

Nephrolithiasis has high recurrence rates, which are about 35–50% following the first renal colic [3].

In this review we summarized the main molecular mechanisms known to date in regard to nephrolithiasis to provide a complete picture and highlight how to make an early diagnosis, in order to reduce the rates of recurrences and the impact of this pathology on patients and every country’s healthcare system.

## 2. Materials and Methods

We performed non-systematic research of the literature in August 2023 and updated it in December of the same year. The literature research was performed on PubMed (MEDLINE) and Web of Science using “Nephrolithiasis” and “Kidney stones” as key words. All the authors agreed on the main articles collected for each topic and a narrative review of the literature was performed.

## 3. Pathophysiology and Main Molecular Mechanisms of Nephrolithiasis

Kidney stones can be broadly categorized into calcareous (calcium-containing) and non-calcareous stones [7]. Calcium-containing stones are radio-opaque and account for almost 75% of cases. They are made of calcium oxalate (CaOx) and/or calcium phosphate (CaP) [7]. Less frequent stones are those made of carbapatite (15.6%), urate (12.4%), struvite (magnesium ammonium phosphate, 2.7%), and brushite (1.7%) [8]. The primary composition of the stones is depicted in Figure 1.

Renal stone formation mechanisms are debated and usually depicted as a step-by-step complex process as illustrated in Figure 2. Such a process starts with urine supersaturation, followed by crystal nucleation, growth, aggregation, and retention in the kidney. However, such events have also been described in non-stone formers [9], validating a multifactorial etiology hypothesis [10]. Recent studies suggest that nephrolithiasis should be regarded as a systemic disease due to the interaction of multiple risk factors [5]. 

The next challenge for researchers is to evaluate how all these risk factors (including diet, gender, and environment) interact with one another and, considering the familiarity with stone formation, the role of genetic predispositions [7].

Body mass index (BMI), fluid intake, calcium intake, and sugar sweetened drink consumption are considered modifiable risk factors for nephrolithiasis [2]. Indeed, a diet high in fructose and obesity with a BMI of 30 or above, especially in a young population, are strongly associated with an increased incidence of nephrolithiasis [7].

Obesity can mediate the occurrence of nephrolithiasis by changing the urinary pH and excreted components [2], with an effect that is differential based on gender [6].

**Figure 1 ijms-25-03075-f001:**
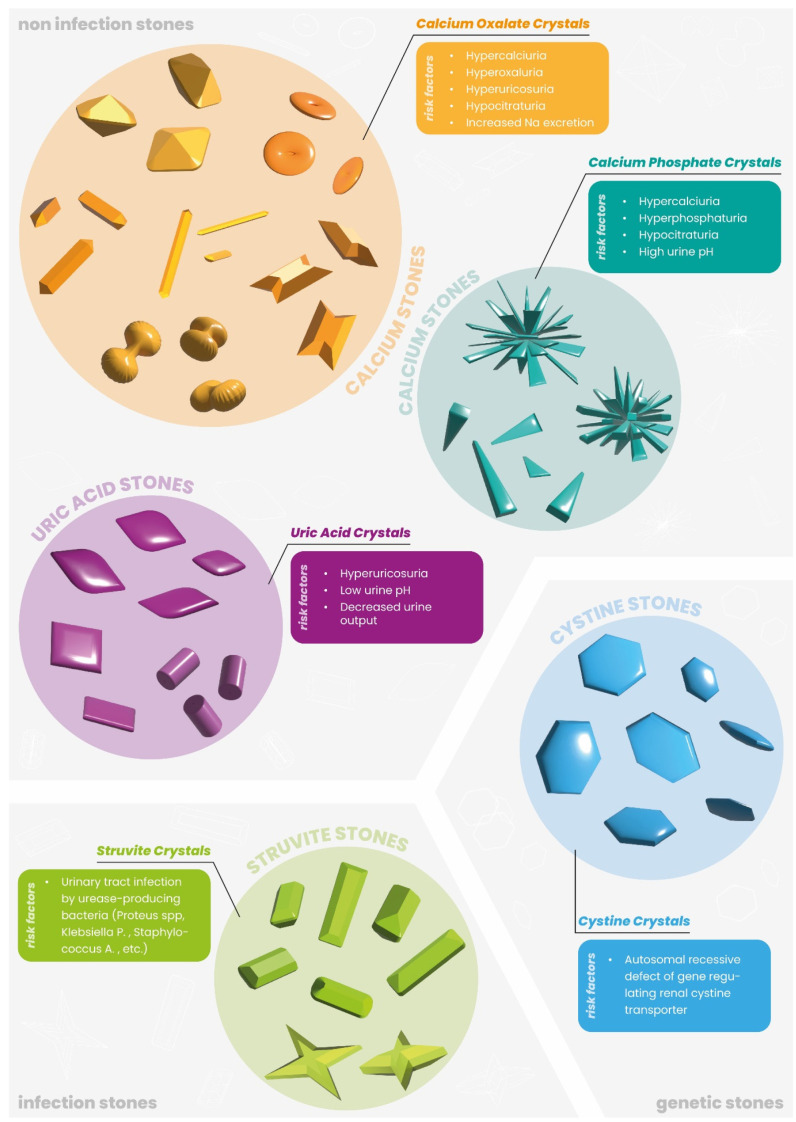
Schematic presentation of main types of kidney stones based on composition. The figure schematically presents the main mechanism of stone formation [11].

**Figure 2 ijms-25-03075-f002:**
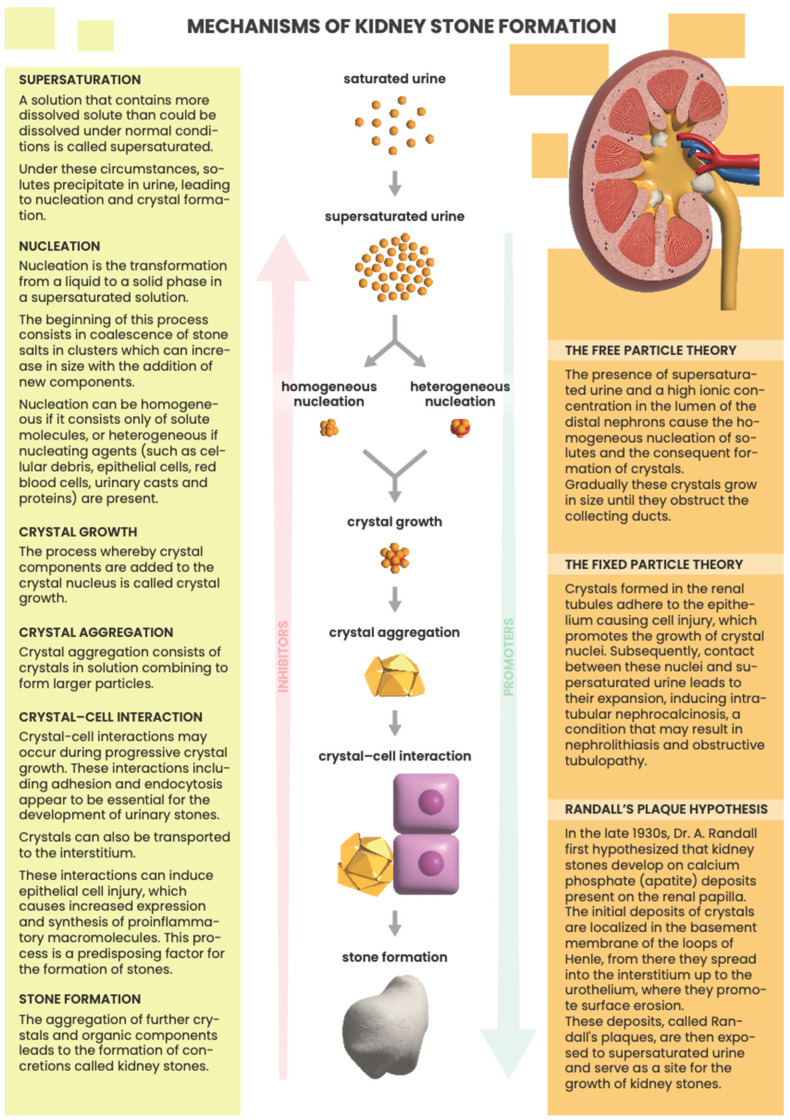
Kidney stone formation: principal theories [12,13,14].

### 3.1. Calcium–Containing Stones 

Calcium-containing stones are mainly composed of CaOX (50%), CaP (5%), or both (45%) [9].

A strict correlation between CaOX stone formation and diet has been described [7]. Higher sodium intake leads to higher natriuresis and urinary excretion of calcium. Similarly, higher oxalate intake might lead to increased oxaluria. Additionally, higher animal protein/sulphated amino acid intake leads to more acidic urine favoring stone formation [7]. 

A vegetarian diet, compared to a normal mixed diet, is associated with increased urinary oxalate excretion [15]. So, a vegetarian diet without adequate calcium intake may not be recommended to patients with mild hyperoxaluria. However, there are nutrition-dependent conditions that are associated with the risk of developing nephrolithiasis, such as insulin resistance, diabetes, obesity, and metabolic syndrome, whose association cannot be mechanistically and fully explained [7].

The differential effect of modifiable risk factors on stone formation has been partially explained by epigenetic modulation [7]. Epigenetic modifications regulate the expression of multiple genes affecting several cellular and metabolic pathways [7]. In a recent study, Khatami et al. showed significant differences in methylated, hypermethylated, and unmethylated status in stone formers vs. controls for genes associated with the vitamin D receptor, calcium-sensing receptor, and claudin 14 [16].

Two main forms of CaOX stones are recognized, namely CaOX monohydrate (COM) and dihydrate (COD), or a combination of both [9]. COM is the most stable form of the crystal and is more frequently observed than COD. Within the internal environment of the human body, the solubility of COD surpasses that of COM. Consequently, the supersaturation of COM consistently exceeds that of COD [17]. A more plausible hypothesis regarding the genesis of such kidney stones posits that during the initial stages of stone formation, COD crystals undergo nucleation and growth [17]. Subsequently, COD gradually undergoes a transformation into COM, starting from the central region of the COD stone. The prior nucleation of COD is a natural occurrence, as high supersaturation levels for both the stable and metastable phases prompt the preferential nucleation of crystals in the metastable phase due to lower interface energy in this phase [17]. Therefore, the transformation from COD to COM primarily involves a solution-mediated process, necessitating the dissolution of COD crystals [17]. However, the supersaturation levels of COM and COD in urine consistently remain high enough for the growth of calcium oxalate crystals. In addition, the surface of a kidney stone remains consistently exposed to urine, creating a highly supersaturated environment, thereby eliminating the possibility of COD and COM dissolution [17]. In the case of a COD kidney stone, numerous COD crystals accumulate, creating multiple gaps between the crystals within the stone [17].

Studies relying on quantitative mass spectrometry showed a large variety of protein encompassed within the COM matrix [18]. It is possible that such proteins stimulate COM aggregation based on polyanion–polycation aggregation based on electrostatic attraction [19]. This hypothesis finds groundwork in previous data that showed how protein aggregates can contribute to CaOx crystal aggregation and nucleation [20,21]. An experimental model evaluated protein aggregation mediated by polyarginine. The comparison between the polyarginine aggregate and the COM stone matrix showed surprising similarity [19]. It is interesting to note that in polyarginine-rich aggregate, polyanions are attracted one by another, while at lower polyarginine concentrations, polyanion attraction is shifted to the supernatant phase [19]. These results suggest a primary role of protein in matrix formation [19]. It is interesting to note that protein-enriched solutions also change the aggregation tendency of polycations such as hemoglobin alpha and histone H4 [19]. In particular, it was noticed that in the presence of a high concentration of polyarginine, its cationic charges cover the negative charges in the urine proteins, displacing potential bindings for cationic proteins, such as hemoglobin alpha or H4, that would be pushed into the supernatant phase [19]. This evidence suggested that the behavior of proteins is likely influenced by heightened electrostatic attraction forces within the polyarginine-rich aggregates [19]. It might be hypothesized that a similar mechanism regulates the formation of the COM matrix. 

Macrophages have long been known to be associated with interstitial crystals in human kidney papillae. Taguchi et al. showed that pro-inflammatory (also termed M1) and anti-inflammatory (termed M2) phenotypes of macrophages are involved in the process of kidney stone formation [22]. M2 macrophages (stimulated by CSF-1, IL-4, and IL-13) can phagocytose CaOx crystals reducing stone development. The signaling mechanisms that promote M2-like macrophage polarization toward CaOx nephrocalcinosis include the NLRP3, PPARg-miR-23-Irf1/Pknox1, miR-93-TLR4/IRF1, and miR-185-5p/CSF1 pathways [22]. Anders et al. investigated the function of NLRP3 in CaOx nephrocalcinosis in NLRP3-deficient mice on a high-oxalate diet [23]. Their findings highlighted that NLRP3 inhibition leads to a shift in infiltrating renal macrophage cells from the M1-like to the M2-like phenotype and the attenuation of renal fibrosis [23]. Proteomic findings have indicated that patients who form kidney stones mainly express M1-like macrophage-related proteins [22] and the microarray analysis of the genes expressed in the kidney papillary tissue of these patients showed the up-regulation of genes associated with the M1 phenotype and a down-regulation of genes linked to the M2 phenotype compared to controls [24]. M1 cells promote stone development through renal inflammation, fibrosis, and cellular damage [22]. 

The importance of inflammation is also suggested by the evidence of a deep correlation between renal tubular epithelial cells (RTECs) and crystals. Such an interaction is crucial and mediates the production of proinflammatory proteins, such as TNF-alpha and IL-1b, through the activation of macrophages. The overproduction of reactive oxygen species (ROS) might work contemporaneously as a cause and a consequence of inflammation in a vicious cycle where tubular injury due to crystal presence promotes inflammation which promotes crystal formation [25]. Khan et al. showed that antioxidant deficits are common in stone formers and are likely a trigger for stone formation owing to oxidative damage to cells [24]. It is suggested that oxidative stress could cause tubular injury which results in crystal retention in renal tubules and stone formation [9]. In this model, ROS and the activation of NLRP3 inflammasomes initiate the NF-kB signal pathway stimulating macrophages to produce inflammatory factors. The consequent inflammatory deregulation leads to CaOX stone formation [25]. This model is supported by Joshi et al. whose results show that in NLRP3 gene-deficient mice who received a high oxalate diet, the formation speed of CaOx stones was significantly lower compared to that of normal mice [26]. 

Clinical studies have confirmed that the levels of malondialdehyde (MDA), thiobarbituric acid reactive substances (TBARS), a-glutathione-s-transferase (a-GST), b-galactosidase (GAL), and N-acetyl-b-D-glucosaminidase (NAG) in the urine of CaOx nephrolithiasis patients are significantly higher than those of healthy patients, suggesting that the development of renal calculi is related to RTEC injury induced by ROS [25]. A high concentration of ROS can induce the excessive activation of autophagy and directly cause cell death and accelerate the progression of an inflammatory response. Moreover, autophagy plays a bidirectional regulatory role in diseases associated with its activation of the NLRP3 inflammasome pathway [25]. 

Vitamin D deficiency has an impact in causing inflammation and oxidative stress [9]. Therefore, it is essential to note that vitamin D deficiency can also contribute to the exacerbation of kidney stone formation or severity [9]. It has been found that vitamin D inhibits the production of pro-inflammatory cytokines and the activation of its receptor (VDR) has been shown to inhibit NF-κB activation [9]. Furthermore, individuals with vitamin D deficiency exhibited increased activity in the Renin–Angiotensin–Aldosterone System (RAAS) and elevated levels of Angiotensin II. From studies conducted in vitro and on animal models, it has been shown that the activation of the vitamin D receptor (VDR) by vitamin D inhibits the transcription of the renin gene, potentially leading to the suppression of oxidative stress [9]. In an expected way, vitamin D deficiency was shown to be highly prevalent among kidney stone formers [9]. Moreover, some studies found that the incidence of vitamin D deficiency was higher in stone formers than in non-stone formers [9].

Primary hyperparathyroidism (PHPT) is a common endocrine disease characterized by hypercalcemia and high or inappropriately normal levels of parathyroid hormone (PTH). Renal involvement is a major complication of PHPT, and silent nephrolithiasis occurs even in asymptomatic patients, with a prevalence ranging between 7 and 35% [27]. Hypercalciuria is a well-established risk factor for nephrolithiasis in patients with and without PHPT, and a continuous relationship between urinary calcium and the risk of nephrolithiasis has been demonstrated in patients with kidney stones [27]. European Association of Urology (EAU) guidelines on urolithiasis have recently indicated the value of urinary magnesium of <72 mg/24 h as the threshold for an increased risk of nephrolithiasis in non PHPT subjects [27]. A recent study by Saponaro et al. found that hypomagnesuria and the urinary Ca/Mg ratio, as well as hypercalciuria, are associated with silent nephrolithiasis in a cohort of patients with asymptomatic sporadic PHPT [27]. Furthermore, they noticed that hypomagnesuria and the urinary calcium/magnesium ratio had a higher sensitivity and a better, but still rather low, positive predictive value (PPV) compared to hypercalciuria (>400 mg/24-h) by itself [27].

Calcium deposition in the kidneys can either lead to nephrocalcinosis when it happens in renal parenchyma, or to the formation of calcium kidney stones when it happens in renal tubules. Various pieces of evidence, including human renal tissue biopsies and intra-operative endourologic imaging, have led to the development of several mechanisms aimed at elucidating the initiation and progression of the stone formation process in the urinary tract. Among the most considered mechanisms in this context is the overgrowth on interstitial CaP plaques (Randall’s plaque), and every mechanism probably contributes to this process [9] (Figure 3).

Randall’s plaques are interstitial plaques composed of CaP deposits mixed with collagen fibers, membrane-bound vesicles, some unidentified fibrillary material, and other products of cellular degradation [28]. Detailed endoscopic examination of kidneys from stone formers suggests that these can be detected in the basement membrane, despite the fact that interstitial plaques are common in both stone formers and non-stone formers and not all plaques are associated with kidney stones [24]. 

In most cases, nephrocalcinosis affects the renal medulla rather than the cortex. The primary risk factor associated with nephrocalcinosis is an elevated urinary excretion of calcium. Additionally, heightened urinary excretion of phosphate and oxalate represents another common risk factor for nephrocalcinosis [9]. Although calcium excretion seems to have a central role in plaque formation, some studies in animal models indicate that both hyperoxaluria and hypercalciuria lead to the formation of intratubular deposits, but not interstitial deposits [9]. This means that additional biological factors, such as deficiencies in crystallization inhibitors, seem to be required for the induction of interstitial crystal deposition [24]. According to this, from the analysis conducted in mice with reduced or absent expression of mineralization inhibitors, such as uromodulin and/or osteopontin, CaP crystal deposits were found [24]. Moreover, genome-wide gene expression profiling of between non-Randall’s plaque and Randall’s plaque tissues showed that immune responses, inflammation, and production of ROS are connected to the formation of Randall’s plaques and idiopathic stones [24]. There is an up-regulation of pro-inflammatory genes (such as *IL11* and *LCN2*) and a down-regulation of anti-inflammatory genes (such as *NACLN*) in the kidney papillary tissues with plaques [24]. In addition, these osteogenic changes, which lead to the deposit of membrane-bound vesicles into the basement membrane, are showed to be connected to the formation of Randall’s plaques and idiopathic stones [24]. However, these aspects need to be investigated further in order to have a clear answer and develop new target therapies. 

In recent years, the microbiome has been shown to be involved in maintaining homeostasis and the pathological processes of diseases [29]. Emerging evidence indicates the active participation of the gut/microbiome in the pathogenesis of nephrolithiasis. Oxalate is excreted via the urine after absorption in the intestine [30]. The lack of commensal bacteria with oxalate-degrading activity has been shown to be associated with stone formation. Observations have shown that the overall microbial composition in patients with kidney stones is considerably different from that in healthy controls, which further supports the intestinal microbiota as an important contributor to stone formation [30]. Karstens et al. have shown the presence of a urinary microbiome in healthy individuals [31]. The urinary microbiome bears a stronger correlation with kidney stone disease than the gut microbiome [32]. *Escherichia coli* as well as *Staphylococcus* have been identified as predictors of nephrolithiasis [33]. Such microbes produce enzymes, such as urease, which increase urinary pH [29]. The study of Gao et al. confirmed the role of these two, but, for the first time, demonstrated that *Mycoplasma* and *Micrococcus* also have a role [29]. 

### 3.2. Non-Calcium-Containing Stones 

Recent epidemiologic studies have highlighted the increasing incidence of uric acid nephrolithiasis in the United States and globally [34]. The proportion of uric acid stone formers increased significantly over the past 30 years from 7 to 14% [35]. In humans, purine undergoes a process of digestion in which uric acid is the main product. Purine can originate from three primary sources: (1) cellular RNA generated through cell turnover, (2) metabolic hepatic synthesis, and (3) dietary consumption of high-purine foods [36]. In particular, purine breakdown results in xanthine, which, under the enzymatic influence of xanthine oxidase, transforms into uric acid [36]. Although xanthine is more soluble than uric acid, elevated levels can also lead to stone formation and nephropathy [36]. 

Endogenous uric acid synthesis remains relatively constant at approximately 300 mg-400 mg per day [36]. Dietary factors, particularly a high-purine diet, can substantially elevate urinary uric acid excretion by 50% or more. However, dietary contributions may vary, with diet accounting for about 50% or less of total daily uric acid production [36]. 

Acidic urine, with or without low ammonia levels, represents the main risk factor for uric acid stone formation [36]. Additional risk factors include hyperuricosuria and low urine volumes [36]. Acidic urine, hyperuricosuria, and low urine volumes could act alone or in different combinations. It noteworthy that hyperuricosuria also increases the risk of CaOX urolithiasis [34].

Acidic urine is recorded in those with impaired ammonia genesis or with increased net acid excretion (NAE) [34]. Interestingly, insulin leads to increased production of ammonia in the proximal tubules and, subsequently, the Na+/H+ ion exchanger [37]. Consecutively, the urine pH is lower, thanks to the reduction in both the production and transport of ammonia caused by the insulin resistance [37]. Insulin resistance is both one of the parameters, together with being overweight, of metabolic syndrome (MS) and the most important factor linked to stone formation in patients with MS [37]. Strohmaier et al. enrolled patients with pure uric acid nephrolithiasis in order to investigate how being overweight could be involved in the formation of uric acid stones [37]. In particular, in their series, the mean BMI was 30.0 kg/m^2^. [37] Confirming the participation of multiple risk factors, the correlation between urine pH (*p* = 0.40), being overweight (*p* = 0.16), and insulin resistance (*p* = 0.12) was not significant [37]. Interestingly, also in non-stone formers, being overweight and obesity are positively correlated with higher NAE associated with a higher fraction of acid excreted as ammonium (NH4⁺/NAE) [34]. 

Struvite or infection stones are quite common and generally pose a difficult treatment dilemma. The presence of urinary infection with a urease-producing organism is necessary for these stones to form [38]. Infection urinary stones resulting from urease-producing bacteria are composed of struvite and/or carbonate apatite. Bacterial urease splits urea and promotes the formation of ammonia and carbon dioxide leading to urine alkalinization and the formation of phosphate salts. Proteus species are urease-producers, whereas a limited number of strains of other Gram-negative and positive species may produce urease. Ureaplasma urealyticum and Corynebacterium urealyticum are urease-producers that are not isolated by conventional urine cultures but require specific tests for identification [39]. Factors that may predispose one to urinary tract infections increase the likelihood of struvite stone formation [38].

Cystine stones are infrequent, representing a small percentage of kidney stones in both adults and children. Cystine stones result from inactivating mutations in genes that encode renal tubular transporters that reabsorb the amino acid cysteine. The 90-kd type II glycoprotein is responsible for the reabsorption of cystine. The human gene for this protein is gene *SLC3A1* and it resides on chromosome 2. The most frequent mutations that were found involved the substitution of the threonine for methionine 467 [40]. This results in an excessive concentration of cysteine in the urine, facilitating the formation of cystine crystals and eventually stones.

## 4. Diagnosis 

The diagnosis of nephrolithiasis involves a thorough assessment, including a detailed medical history and physical examination, consideration of the patient’s risk factors and concurrent medical conditions, and an evaluation of the probability of a significant alternative diagnosis. 

The diagnosis varies depending on the type of calculus. Typically, laboratory tests are conducted, encompassing hematological and urinary analyses, which prove invaluable in identifying the causative factors associated with the formation of distinct urinary stone types as suggested by EAU guidelines 2023 (Table 1).

For the diagnosis of symptomatic kidney stones, it is necessary to visualize the actual stone either during surgical resection or after urination using imaging techniques. CT is the optimal imaging modality for stone detection and is superior to ultrasonography and plain abdominal radiographs in detecting small or distal ureteral stones [42]. However, it is sometimes impossible to make a definitive diagnosis of symptomatic kidney stones because the patient’s symptoms may abate and the stone may go away without being noticed [42]. Ultrasound represents the best investigation of first instance in the diagnosis of urinary stones with a sensitivity value of 61% and a specificity close to 100% [42]. Spiral CT, without the injection of a contrast medium, is now widely used as the method of choice in the emergency diagnosis of patients with renal colic, with equivocal or negative ultrasound and/or radiographic findings, with a sensitivity of 97% and a specificity of 96% [43].

Ultrasound has a sensitivity of 45% and a specificity of 94% for ureteral stones and a sensitivity of 45% and a specificity of 88% for renal stones [44,45].

Ultrasound often misses stones that are visible on CT; even when stones are identified through ultrasound, this imaging method tends to overestimate stone size compared to CT scans. Small stones initially visible on CT may be overlooked in subsequent ultrasound examinations due to reduced spatial resolution [42]. 

For calcium-containing calculi, they are clearly visible on CT without contrast Direct abdominal X-ray is no longer used as a first-line diagnosis method, but it still shows the lithic formation in the renal shadow. At the laboratory level, blood and urine analysis takes specific parameters into consideration.

Among non-calcium-containing stones, cystine stones are discernible on standard X-rays due to the sulfur content of cysteine, but even with ultrasound and CT [40].

Non-contrast CT and ultrasound can identify uric acid stones, which appear as radiolucent on a plain abdominal X-ray [46]. 

The current guidelines from both the EAU and the recently issued AUA guidelines now include a suggestion for stone analysis in individuals forming stones for the first time, as well as in some recurrent high-risk stone formers [47,48]. 

Standard techniques for stone analysis include polarization microscopy on grain preparations, chemical methods using analysis kits (CA), and other contemporary approaches, like X-ray diffraction (XRD) and infrared spectroscopy, with the rapid FTIR technique being particularly noteworthy. Presently, the favored analytical methods are infrared spectroscopy and X-ray diffraction (XRD) [49,50]. 

However, although the utilization of CA for stone composition is diminishing, it continues to be employed globally due to budget constraints and a lack of adequate awareness regarding its limitations.

Understanding the composition of stones plays a crucial role in individuals forming non-calcium stones. For example, identifying a stone as pure cystine or detecting even a minimal percentage of cystine serves as a diagnostic indicator for cystinuria, an inherited autosomal recessive disease. Moreover, the identification of a composite stone consisting of struvite and CaOx strongly suggests the presence of an underlying metabolic disorder [51].

Conversely, in calcium stones, the precise identification of the mineral components holds significantly less relevance. The management of kidney stones may require a multidisciplinary approach involving physicians, radiologists, and urine specialists.

## 5. Recurrences of Nephrolithiasis 

Nephrolithiasis is characterized by a high risk of recurrence. The spontaneous 5-year recurrence rate is 35% to 50% following the first renal colic [3]. Furthermore, if patients do not apply metaphylaxis, the relapse rate of secondary stone formation is estimated to be 10–23% per year, 50% in 5–10 years, and 75% in 20 years from the first occurrence [3] Approximately more than 10% of patients could experience more relapses [52]. Additionally, the recurrence rate of urinary calculi in patients with specific stone mineral compositions and morphologies can even be up to 82.4% [52].

Recurrences can exhibit either as asymptomatic alterations in kidney stone burden detected through imaging, or as symptomatic episodes caused by the migration of stones. 

Every patient has a different frequency of kidney stone symptom recurrence: some individuals can undergo only a single isolated event, while others have to face frequent recurrences with chronic pain and necessitating multiple surgeries [53]. Successive studies, like radiographic imaging, are necessary in order to detect recurrence and to identify alterations in kidney stone burden, distinguishing new stone formation, stone growth, and the passage of pre-existing stones. 

Over a 5-year follow-up period, nearly 19% of patients experienced symptomatic recurrence necessitating clinical care. However, this datum can vary when self-managed symptomatic recurrence was taken into account, with a rate of close to 30% [54]. A kidney stone may develop, enlarge, and be expelled either with or without noticeable symptoms. In fact, approximately half of the asymptomatic kidney stones found in individuals experiencing their first symptomatic stone episode will naturally pass within a span of 5 years. Additionally, about half of these individuals will encounter symptoms during the process of stone passage [54]. The 5-year recurrence rate for individuals experiencing their first symptomatic stone episode rose from 30% to 67% when both radiographic and symptomatic recurrence were taken into account, as opposed to considering only symptomatic recurrence [54]. At 5 years, 35% of patients formed a new stone, 24% had enlargement of a retained stone, and 26% passed a previously retained stone [54].

There is evidence that having more past stone episodes predicts an increased risk of future episodes [54]. Among incident symptomatic stone formers, the percentage that will have at least one concurrent asymptomatic stone at baseline is about 50%, and half of these stone passages will be caused a symptomatic episode [54]. For individuals experiencing their first symptomatic kidney stone episode in the community, the recurrence rates for confirmed symptomatic episodes requiring clinical care varies according to the temporal distance from the first episode: 11% at 2 years, 20% at 5 years, 31% at 10 years, and 39% at 15 years [54]. Furthermore, after each subsequent kidney stone episode, the recurrence rates tend to increase. Specifically, the estimated 5-year recurrence rates were 17%, 32%, 47%, and 60% following the first, second, third, and fourth or higher episodes, respectively [54].

## 6. Risk Factors and Prevention of Nephrolithiasis

A prediction tool for the risk of a second kidney stone episode is needed to optimize treatment strategies. Identifying patients who are at high or low risk for symptomatic recurrence can help make informed decisions regarding lifetime commitment to various stone prevention interventions [55]. The Recurrence of Kidney Stone (ROKS) model predicts the risk of a second symptomatic kidney stone episode after the first episode based on 11 predictors [55]. The model was developed from 2239 first-time adult renal stone formers with evidence of a past, obstructive, or infected stone causing pain or gross hematuria from 1984 to 2003 [55]. The presence of a family history of nephrolithiasis and a prior suspected stone episode predict stone recurrences [55]. However, this tool was not applicable for predicting recurrence in those who have had two or more symptomatic episodes [55].

After 4 years, the ROKS nomogram was revised to predict the risk of symptomatic recurrence after each stone episode in a population-based cohort of symptomatic kidney stone formers. A total of 27 candidate predictors at the first and subsequent stone episodes, the number of past episodes, and the recurrence-free time interval since the last episode were evaluated to predict symptomatic recurrence. The researchers also investigated 24 h urine chemistries among the subset of patients with available laboratory measurements as predictors of recurrence [55]. Since it has been seen that specific dietary habits (Mediterranean diet for instance) act as a protective factor against nephrolithiasis [7], the researchers further assessed the prevalence of dietary, medical, and surgical interventions with increasing number of stone episodes and assessed any effect of these interventions on the risk of symptomatic recurrence [55]. That study also identified novel risk factors for stone recurrence, including the number of past stone episodes and years since the last episode [55].

The revised ROKS prediction tool for estimating symptomatic recurrence risks can be used to guide the management of stone formers by individualizing preventative interventions based on the risk of symptomatic recurrence. An electronic version of the ROKS nomogram is available on the QxMD app “Calculate” (iOS: http://qx.md/qx; Android: http://qx.md/android; and web tool: http://qxmd.com/ROKS; accessed on 23 January 2024) [55] This tool can also be used to estimate symptomatic stone episode rates that result in clinical care for use in clinical trials on kidney stone prevention. If the subset at high risk for another episode can be identified, subspecialty evaluation with 24 h urine chemistries, radiographic monitoring for stones, medical therapy, or more intensive dietary counseling may be of benefit [55]. Early effective interventions may spare such individuals the morbidity of painful stone episodes and potential long-term complications, such as kidney failure [55].

Current guidelines recommend maintaining a urine output >2.0–2.5 L/day and a fluid intake of 2.5–3.0 L/day [56]. Some beverages consumed every day, such as coffee and tea, can also exert a diuretic effect. 

To prevent kidney stones from forming, it is important to discover possible metabolic pathologies, such as primary hyperparathyroidism. Several metabolic diseases, such as hypercalciuria, hyperoxaluria, hypocitraturia, hypomagnesuria, and hyperuricemia, are known risk factors for kidney stone formation. It is possible to identify metabolic risk factors with urine analysis and an examination of the composition of the stone.

In addition, weight loss in those with obesity or who are overweight is also recommended in order to maintain a normal body mass index (BMI) to reduce the risk of kidney stone formation [56]. An in vivo study also showed that weight loss through dietary restriction and exercise increases urinary citrate excretion and reduces the risk of kidney stone disease [56]. In both sexes, physical activity was inversely associated with nephrolithiasis prevalence. In addition, athletes have some protective factors, such as high levels of magnesium in the urine. Fluid balance during and after strenuous physical activity and exercise also needs to be considered, which can lead to profuse sweating and dehydration [56].

In people with arterial hypertension, it is strongly recommended to reduce salt intake to no more than 5 g of table salt [56]. Restricting sodium intake, combined with higher intakes of potassium, magnesium, and citrate, not only reduces the risk factors for urinary stone formation, but also prevents bone loss and hypertension, despite blood flow, kinetic sensitivity to sodium increases intake and reduces urolithiasis. Renal function in older adults must also be considered [7,56].

Fruits and vegetables might contain substances that can be harmful to someone who develops stones, in particular, chard or spinach contain large quantities of calcium oxalate and oxalate stones are frequent in the producers of this type of stones [56]. Still, many fruits and vegetables can increase the levels of citrate in the urine, which can inhibit stone formation [56]. A Mediterranean diet (defined as high consumption of plant-derived foods with monounsaturated to saturated fatty acids and low consumption of meats), is associated with a lower risk of kidney stone disease [7].

It is well known that a diet rich in meaty animal protein must be limited and individualized, as changes in urine acidity can lead to the formation of many types of stones, mainly those favoring uric acid proteins that are released, which favor the formation of urate stones [52].

Some prior studies also reported a reduced risk of nephrolithiasis in patients who consumed alcohol [2]. In addition, the protective effect of alcohol consumption on nephrolithiasis showed a dose-dependent association, with a 10% decrease in the rate of nephrolithiasis for 10 g/day of alcohol consumption [2]. However, there was no protective effect of alcohol consumption on nephrolithiasis when it was excessively consumed [2]. 

Risk factors, including younger age, a higher BMI, Caucasian race, family history of nephrolithiasis, personal history of nephrolithiasis, suspected nephrolithiasis episode prior to first confirmed stone episode, any concurrent asymptomatic (non-obstructing) stone, hypertension, uric acid stone, pelvic or lower pole nephrolithiasis, surgery, and 24 h urine test completion were identified to be associated with the relapse of kidney stone disease [52]. 

Thiazide diuretics, allopurinol, and citrate supplementation are effective in preventing calcium stones that recur despite lifestyle modification, even in the absence of hyperuricemia, urinary acidosis, hypocitraturia, or hyperuricosuria [10]. Thiazides are most often prescribed to treat arterial hypertension but, in addition to their blood-pressure-lowering diuretic effect, they also significantly affect renal calcium handling. Thiazide treatment results in hypocalciuria, a property used to treat hypercalciuria in calcium kidney stone formers [57].

The effectiveness of thiazide diuretics has been documented only with high dosages (e.g., hydrochlorothiazide, 50 mg per day; chlorthalidone, 25 to 50 mg per day; indapamide, 2.5 mg per day); lower dosages have fewer adverse effects, but their effectiveness is unknown [10].

However, recently, Dhayat et al. performed a randomized trial to assess the efficacy of various doses of hydrochlorothiazide in preventing the recurrence of kidney stones [58]. 

The primary goal of this trial was to explore the dose–response relationship concerning the primary endpoint, which comprised a combination of the symptomatic or radiologic recurrence of kidney stones. In this randomized, double-blind trial, patients experiencing recurrent kidney stones containing calcium were enrolled and randomly allocated to receive hydrochlorothiazide at a daily dose of 12.5 mg, 25 mg, or 50 mg, or a placebo once daily [58]. 

The participants were monitored for a median duration of 2.9 years [58]. Eventually, they demonstrated that the recurrence rates were comparable among the hydrochlorothiazide groups receiving doses of 12.5 mg, 25 mg, and 50 mg, as well as the placebo group [58]. Therefore, these findings raise concerns regarding the prolonged use of hydrochlorothiazide for kidney stone prevention [58]. 

Allopurinol should be started at 100 mg once per day and increased gradually to 100 mg three times per day [10]. There is no evidence that combination therapy with thiazide diuretics or alkaline citrates is more effective than monotherapy [10]. Allopurinol is one of the mainstays of therapy for patients with calcium stones, but most patients with uric acid stones have acidic urine that requires treatment with alkaline citrates [10]. Citrate supplements are not only used for calcium stones, but also for uric acid stones (with a urine pH target of 6.0 to 7.5 or higher) and cystine stones (with a urine pH target of 7.0 to 7.5 or higher) [10]. The preferred dietary salt supplement is potassium citrate, with a target dose of 5 to 12 g per day [10]. The starting dose should be 9 g per day, divided into three doses and taken within 30 min of a meal before bed—a snack. The most common side effects are gastrointestinal problems. Potassium citrate supplementation can correct low serum potassium levels caused by thiazide diuretics, but there is no evidence that combination therapy is more effective than either drug alone [10]. Sodium citrate is an alternative to citrate supplementation, but the resulting excretion of sodium and calcium may partially offset the desired effect. Unsweetened soda is a tastier and less expensive alternative to citrate supplements. Although there is no direct evidence of its effectiveness in preventing stone recurrence, the dilution of lemon juice in water should help patients meet the recommended fluid intake [10].

If medication or citrate supplementation is prescribed, serum potassium levels (for patients taking thiazide diuretics or potassium citrate) and liver enzymes (allopurinol) should be monitored to detect potentially serious adverse effects [10]. Potassium levels should be monitored before prescription, within two weeks of prescription, and then every 12 months (earlier if illness occurs or another medication is added) [10].

Urinary acidification has been proposed for the prophylaxis of infection stones, but long-term acidification is difficult to achieve in urine infected by urease-producing bacteria. Urease inhibitors lead to the prevention and/or dissolution of stones and encrustations in patients with infection by urea-splitting bacteria, but their use is limited by their toxicity [39]. Furthermore, the administration of citrate salts involves an increase in the value of nucleation pH (pHn) [39]. This is the pH value at which calcium and magnesium phosphate crystallization occurs, in a greater way than the corresponding increase in the urinary pH due to its alkalinizing effect and resulting in a reduction of the risk of struvite crystallization [39].

It has been seen that Pioglitazone protects against CaOx crystals by promoting M2-like Mj polarization and decreasing renal CaOx crystal deposition and inflammatory damage in murine bone marrow-derived Mj in vitro and in CaOx nephrocalcinosis mouse models in vivo [22].

Previous studies demonstrated that curcumin (an antioxidant) alleviated COM crystal deposition in the mouse kidney through antioxidant, anti-inflammatory, and antifibrotic activities [59]. In addition to this, a new study was carried out about the role of vitexin in the prevention of nephrolithiasis. The mechanism underlying the protective effect of vitexin involved anti-pyroptotic, anti-inflammatory, antioxidative, and EMT suppressive activities. The results of Ding et al. confirm the protective role of vitexin in nephrolithiasis through the attenuation of NLRP3 activation and the GSDMD-related pyroptosis form of programmed cell death [59]. Apoptosis mediates nephrolithiasis and is activated when renal epithelial cells are exposed to oxalate; the results demonstrated that apoptosis induction in mice with nephrolithiasis and COM crystal-treated HK-2 cells was repressed by vitexin treatment and that the viability of HK-2 cells and THP-1-derived macrophages. which was reduced by treatment with COM crystals, was protected by vitexin in a concentration-dependent manner [59]. In that study, vitexin decreased the levels of M1-type macrophage markers (TNF-α and IL-1β), and these results might indicate that vitexin mitigates macrophage-mediated renal injury by inhibiting NLRP3 inflammasome activation and GSDMD-related pyroptosis along with the expression of inflammatory factors such as TNF-α and IL-1β. In summary, the results of that study indicated that vitexin treatment alleviated crystal deposition and renal tissue injury and could be a promising therapeutic antioxidant [59]. In conclusion, the prevention of the kidney stones can be achieved by taking an integrated approach tailored to a single patient and divided according to the different types of stones (Table 2).

## 7. Future Perspectives 

The study of epigenetic modifications may find the link between genetic predisposition and the environmental factors that are associated with nephrolithiasis. Since Khatami et al. [16] found the presence of epigenetic differences between stone formers and non-stone formers, they inspired an in-depth analysis of this topic in order to unravel the underlying epigenetic mechanism [7]. 

Moreover, recent studies highlighted the correlation of the urinary microbiome with kidney stone disease [32]. As the study of Gao et al. demonstrated, different new microorganisms are related to kidney stone disease, and it would be useful to deepen the investigation of this aspect [29]. 

The research by Srirangapatanam et al. put core body temperature among the factors that could influence stone formation. In particular, they highlighted an unexpected theory against the Ostwald ripening phenomenon [68].

Ostwald ripening is a phenomenon that is inversely dependent on temperature, in which supersaturated molecules in a solution will condense so that smaller particles merge into larger particles and has been well described in human nephrolithiasis [68,69]. According to this, it was expected that patients with nephrolithiasis would tend to have a lower core body temperature [68]. It was seen that ambient temperature and local climate appear to play a role in stone formation, as geographical areas with warmer climates tend to have a higher incidence of stone disease, and presentations to the ER tend to increase after particularly hot days [70,71,72,73]. Srirangapatanam et al., in a retrospective matched case–control study, analyzed the role of core body temperature in the pathophysiological process of stone formation [68]. This study highlighted that patients with a history of kidney stones were noted to have a slightly higher mean oral temperature when compared to matched controls [68]. In particular, each 1 °C increase in core body temperature predicts an average 21% increase in the history of kidney stone formation [68]. Furthermore, with logistic regression, a higher core body temperature was predictive of a history of kidney stones [68]. In this scenario, thermo-responsive proteins, including transient receptor potential channels, are of particular interest. These could undergo conformational changes in response to very small temperature changes, in order to increase the likelihood of either the deposition of Randall’s plaque or the formation of renal calculi [68]. The discoveries that were made by that study lay the groundwork for future studies and interventions in stone prevention [68]. 

Despite these notions, how the core body temperature exactly influences stone pathogenesis remains unclear [68]. Therefore, further studies might discover how it works exactly. Moreover, in this review we discussed the use of drugs to prevent recurrences and, in particular, the findings about the prolonged use of hydrochlorothiazide for kidney stone prevention, but future studies are required to fully standardize this as a therapy. Among the possible future therapeutic strategies, the modulation of microbiota can be mentioned [29]. Since single microbial strains could not sufficiently lower the pathological risk of oxalate metabolism, the strategy could be treatment with a combination of diet control and fecal transplantation [29]. However, before standardizing this as a novel therapy, further studies are required.

## 8. Conclusions

There are several mechanisms behind the process of stone formation, and most of them are not completely known. In this review, we underlined some uncommon factors that are involved in kidney stone formation, such as the role of inflammation and core body temperature. Moreover, we provided some advice, such as the use of the ROKS nomogram and the consumption of substances and drugs in order to prevent recurrences and improve the patient’s quality of life. However, many aspects of nephrolithiasis are not completely known, so ongoing in-depth investigations are essential to gain a deeper understanding in the formation of kidney stones. This is crucial for the development of innovative preventive and therapeutic strategies.

## Figures and Tables

**Figure 3 ijms-25-03075-f003:**
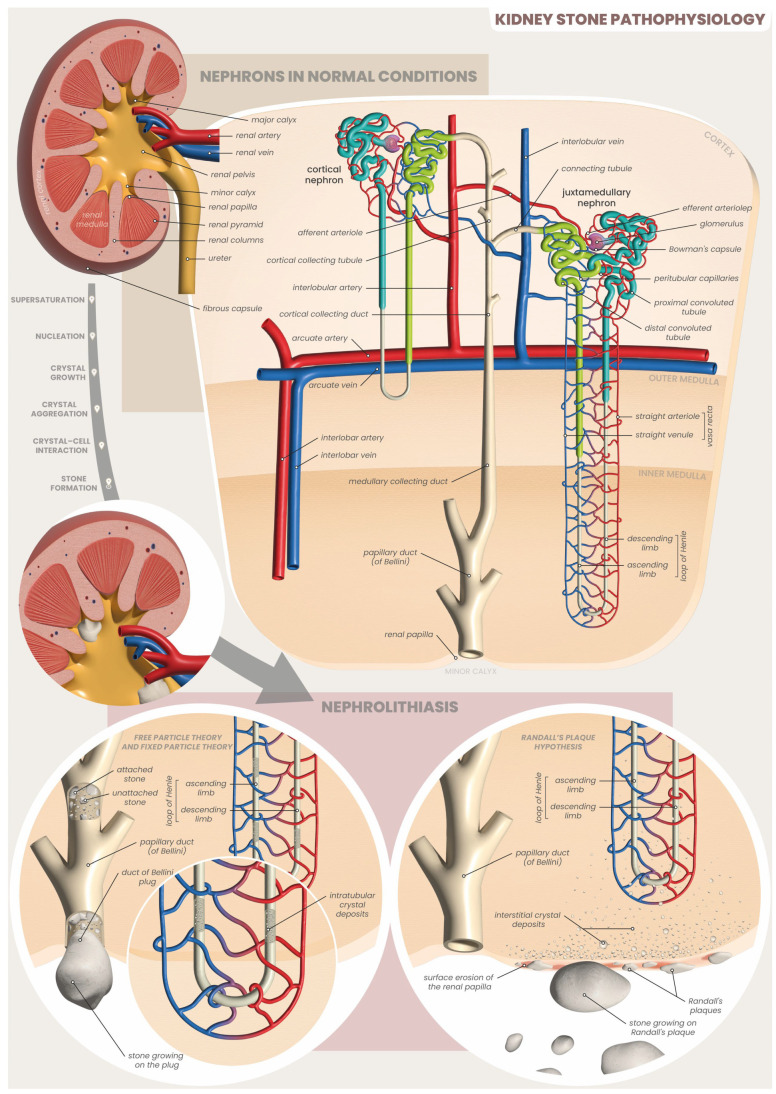
Renal structure and the formation of kidney stones [12,13].

**Table 1 ijms-25-03075-t001:** Recommended blood analysis and urinalysis based on the specific type of stones as suggested by the EAU Guidelines on Urolithiasis. European Association of Urology 2023 [41].

Type of Stones	Blood Analysis	Urinalysis
Calcium Oxalate stones	Creatinine, sodium, potassium, chloride, ionised calcium (or total calcium + albumin), phosphate, uric acid; In the case of increased calcium levels, parathyroid hormone (PTH), vitamin D	Urine volume, urine pH profile, specific weight, calcium, oxalate, uric acid, citrate, sodium, magnesium
Calcium Phosfate stones	Creatinine, sodium, potassium, chloride, ionised calcium (or total calcium + albumin), phosphate, uric acid; In the case of increased calcium levels, parathyroid hormone (PTH) and vitamin D	Urine volume, urine pH profile, specific weight, calcium, oxalate, uric acid, citrate, phosfate
Nephrocalcinosis	PTH (in the case of increased calcium levels), vitamin D and metabolites, vitamin A, sodium, potassium, magnesium, chloride, and bicarbonate	Urine pH profile at different times of the day daily urine volume, specific weight of urine, and levels of calcium, oxalate, phosphate, uric acid, magnesium, and citrate
Uric acid stones	Creatinine, potassium, uric acid levels	Urine volume, urine pH profile, specific weight of urine, and uric acid level
Struvite and infection stones	Creatinine	Urine pH measurements and urine culture
Cystine stones	Creatinine	Urine volume, pH profile, specific weight, and cystine

**Table 2 ijms-25-03075-t002:** Recommended medication based on the specific types of stones as suggested by the EAU Guidelines on Urolithiasis. European Association of Urology 2023.

Type of Stones	Suggested Medication
Calcium Oxalate stones	Fluid intake, diet - Hyperoxaluria → foods with low oxalate content and benefits with Calcium (1000 to 2000 mg/d depending on oxalate excretion) and magnesium [60,61] - Hypercalciuria 5–8 mmol/d → Alkaline citrate or sodium bicarbonate - Hypercalciuria > 8 mmol/d → Hydrochloro-thiazide initially 25 mg/d up to 50 mg/d chlorthalidone 25 mg/d indapamide 2.5 mg/d [62]
Calcium Phosphate stones	Primary hyperparathyroidism → surgery Renal tubular acidosis → bicarbonate or alkaline citrate therapy [60]
Uric acid stones	Fluid intake, diet Hyperuricosuria → purine reduction in their daily diet. Alkaline citrate or sodium bicarbonate plus/or allopurinol [63] - Hyperuricosuria > 4.0 mmol/d → Allopurinol 100 mg/d - Hyperuricosuria > 4.0 mmol/d or hyperuricaemia > 380 μmol → Allopurinol 100–300 mg/d [63,64]
Struvite and infection stones	Fluid intake, diet, complete surgical stone removal, short- or long-term antibiotic treatment, urinary acidification using methionine or ammonium chloride, and advice to restrict intake of urease [65]
Cystine stones	A 24 h urine volume of >3 L, a diet low in methionine, Avoidance of sodium consumption > 2 g/day, Tiopronin, Captopril [66,67]

## Data Availability

Not applicable.

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
