# Peer review of "Pathophysiology and Main Molecular Mechanisms of Urinary Stone Formation and Recurrence"

_ijms, 2024, doi:10.3390/ijms25053075_

Round 1

Reviewer 1 Report

Comments and Suggestions for Authors

This is a well-written review on the pathogenesis of Urolithiasis.

Some comments:

Calcium  oxalate stones: The two types of stones should be mentioned (whewellite (caclium oxalate monohydrate) and weddellite (calcium oxalate dihydrate).

Cystine stone formation should be discussed and not only be menrtioned in the figure.

Uric acid stones: TGhe authors wrote: "Three main factors are involved in the pathogenesis of uric acid nephrolithiasis: 258 low urine pH, hyperuricosuria, and low urine volume."

This statement is correct. However, the typical risk factors could be found in only about two thirds of uric acid stone formers. Furthermore, there was no significant correlation between BMI and urine pH in this largest series of uric acid stone fromers published. (W.L.STROHMAIER, B.M.WROBEL, G.SCHUBERT (2012) Overweight, insulin resistance and blood pressure (parameters of the metabolic syndrome) in uric acid urolithiasis.  Urol Res 40: 171-175)

Comments on the Quality of English Language

see above

Reviewer 2 Report

Comments and Suggestions for Authors

The present study represents a literature review regarding the pathophysiology of urolithiasis. The article is well-written and the following revisions would enhance its overall quality.

-The authors may report the literature databases that were searched to identify potential articles.

-The results of more animal model studies should be exhibited to shed more light on the molecular mechanisms of stone formation.

-A Figure may be added depicting the main molecular mechanisms implicated in urolithiasis. 

-A recent trial regarding the role of thiazide diuretics (10.1056/NEJMoa2209275) should be discussed.

- A section about the implications for future research should be added.

Comments on the Quality of English Language

Minor editing of English language required

Reviewer 3 Report

Comments and Suggestions for Authors

This review summarizing slowly changing knowledge concerning pathogenesis of kidney stones formation. The chapter molecular mechanisms of nephrolithiasis does not describe 'molecular mechanisms' and it would be better to name is pathogenesis of kidney stones formation. In my opinion all the predisposing factors (e.g. hypocitraturia, hypercalciuria, hyperoxaluria, ..) should be defined and presented in the table.

The chapter Diagnosis ... describe mostly imaging. It would benefit from adding clear presentation of laboratory test in the recurrent disease in relation to stone chemistry.

The chapter Prevention. Please add a table summarizing medication in relation to stone chemistry.

Finally the abstract. It is not representative for the study content.

Comments on the Quality of English Language

There are some repeats - starting from the first sentence of abstract 'disease'

 'This review aims to understand all the molecular mechanisms' rather presents pathomechanisms.

Reviewer 4 Report

Comments and Suggestions for Authors

Overview

The article attempts to explore the molecular mechanism of urinary stones, but the overall logic of the article is chaotic, and the idea is shallow, resulting in the article only being a pile of literature without in-depth exploration and logical connection. Therefore, it gives readers a feeling of obscurity and difficulty to understand, and we believe that the article does not have the quality of publication.

Details

1. Regarding the formation of kidney stones, based on the complexity of structures such as the renal pelvis, the author should include an explanatory diagram of the renal structure in the schematic diagram.

2. The formation of kidney stones such as uric acid may be related to hyperuricemia, while the formation of oxalate stones may be related to higher vegetarian intake. Similar triggers and pathogenesis should be further discussed. The author's writing in this section is insufficient.

3. It seems that the author did not have in-depth logical thinking when writing this article. For example, in the abstract part, the incidence rate of kidney stones is rising year by year, which should be mentioned after the first sentence of the definition of kidney stones "one of the most common diseases of the urinary system", followed by the pathogenesis, etiology, specific types, etc., and finally should be the purpose of this study. The author's entire writing has this issue, and there are no logical connectors, making it difficult for readers to understand what they want to say. We believe that the author's current article is obscure and difficult to understand.

4. The main body of the article appears to be a pile of literature, and the author did not provide a comprehensive review of them, nor did they express their own opinions or propose more extended discussions on these studies.

Comments on the Quality of English Language

English very difficult to understand/incomprehensible.

Round 2

Reviewer 1 Report

Comments and Suggestions for Authors

The authors picked up some suggestions. However, my comments concerning uric acid urolithiasis and the appropriate references were not cinsidered sufficiently.

Comments on the Quality of English Language

see above

Reviewer 2 Report

Comments and Suggestions for Authors

The authors have adequately revised their manuscript; therefore, the current form is acceptable for publication.

Comments on the Quality of English Language

Minor editing of English language required

Author Response

We wished to extend our thanks to the reviewer for their evaluation and we hope that our modifications will be considered satisfactory to warrant the publication of our manuscript. 

Reviewer 4 Report

Comments and Suggestions for Authors

The authors tried to answer all of our comments, and we feel that the authors' team has answered all questions in their entirety, so we consider the current quality of the article ready for publication.

Author Response

We would like to express our gratitude to the reviewer for their evaluation and we hope that our modifications will be considered satisfactory to warrant the publication of our manuscript.